# OpenReview forum: "Knowledge-localized Unlearning for Faithful Forgetting in Language Models"
_ICLR.cc/2025/Conference — Submitted to ICLR 2025_

### Official Review · Reviewer_Mrad · 2024-10-31

**Soundness:** 2
**Presentation:** 3
**Contribution:** 2
**Rating:** 3
**Confidence:** 4

**Summary:**

This paper focuses on the issue of superficial unlearning in language models, which refers to the phenomenon where an unlearning method either fails to erase the interconnected knowledge it should remove or unintentionally erases irrelevant knowledge. To investigate the phenomenon of superficial unlearning, this paper introduces a new benchmark, FaithUnBench, to evaluate unlearning methods in real-world knowledge QA settings. Then, it proposes an unlearning method, which identifies and updates only knowledge-related neurons to achieve faithful unlearning.

**Strengths:**

1.This paper explores a promising direction: the timely issue of removing sensitive or private information from language models.

2.This paper defines the problem of superficial unlearning and constructs a benchmark for a more in-depth analysis and evaluation of unlearning methods.

**Weaknesses:**

1.The choice of evaluation metrics is unreasonable. Why is the UA of all baselines equal to 0.33 (only GA on Gemma2 is 30.30)? Does this suggest that the unlearning dataset is simple enough and more rigorous testing methods need to be designed? For instance, WMDP [1] also uses multiple-choice QA to evaluate the unlearning effect, yet existing unlearning methods struggle to reach the level of random guessing.

2.The implemented baselines are relatively few, and more models (LLaMA3) and unlearning methods (such RMU [1] and NPO [2]) need to be compared.

[1] The WMDP Benchmark: Measuring and Reducing Malicious Use With Unlearning.

[2] Negative Preference Optimization: From Catastrophic Collapse to Effective Unlearning.

**Questions:**

1.What is the form of the data to be removed? Is the Base QA used as the training data? This is easier compared to unlearning the Harry Potter books.

---

> ### Author Response · Authors · 2024-11-20
> **Rebuttal by the Author**
>
> Thank you for your feedback. We will address each of your questions with sincerity.
>
> $\\\\$
>
> ### **1. Details of the evaluation metrics (Weakness # 1)**
>
> **1-1. The rationale for the evaluation results**
>
> Our evaluation framework early stops the unlearning process when UA <= 0.33 (random sampling from three options) is satisfied (Section 5.1). Therefore, it is natural that some algorithms tend to excessively unlearn the given forget set and reach UA score below 0.33.
>
> In contrast, WMDP [1] enforces unlearning of the datasets until a fixed number of training batches (e.g., 150, 300, or 500 iterations) is reached. This rigidity makes WMDP inflexible, often leading models to struggle to achieve even the level of random guessing.
>
> In conclusion, the observed results are derived from the gap in the evaluation processes of the two benchmarks (early stop during iterations vs. fixed iterations).
>
> **1-2. FathUnBench vs. WMDP**
>
> Our paper tackles one of the most challenging scenarios: unlearning real-world knowledge, considering its complex and interconnected nature.
>
> We fully acknowledge that WMDP deals with challenging scenarios, focusing on unlearning knowledge across various professional domains (biosecurity, cybersecurity).
>
> However, existing benchmarks, including WMDP, have not accounted for the interconnectedness of knowledge since they have evaluated unlearning methods only on knowledge to forget and other independent knowledge. **In the "Rebuttal for All Reviewers", we provide an in-depth discussion of the characteristics of our benchmark in comparison to others (at the top).**
>
> $\\\\$
>
> ### **2. More models and baselines (Weakness # 2)**
>
> Our method is a model-agnostic method and can be flexibly adapted to any unlearning method. Accordingly, we applied our method to GA$_{ret}$ and it as the primary comparison target. However, we acknowledge that RMU is one of the most powerful backbone methods among recent unlearning methods; thus, we conducted experiments with RMU and also attempted to apply our method to RMU.
>
> For the RMU experiments, we searched for $\alpha_{RMU} \in \lbrace20, 50, 100, 150, 200, 300\rbrace$ and $lr \in [1e-5, 3e-3]$. We also used $c=20$ and $l=7$, following the implementation details on the original GitHub page [2]. In the KLUE+RMU experiments, we adopted the same settings as the RMU but with a key difference in layer selection for updates. RMU manually selects three lower layers ($l-1$, $l-2$, and $l-3$) for updates, but KLUE+RMU automatically searches for neurons to update. For KLUE settings in KLUE+RMU, we selected $\alpha_{KLUE}=10$ and $p=0.2$, using a higher neuron ratio than KLUE+GA$_{ret}$ since only the first to sixth layers are updated
>
> We also adopted NPO [3] with searching for hyperparameters of $\beta \in [0.1, 0.5]$ and $lr \in [5e-6, 1e-4]$. To demonstrate the scalability of our method, we also conducted experiments on the Llama 3.2 (3B) model as a new backbone.
>
> **Gemma-2 (2B)**
>
> |Method|UA|UA$^{‡}$|TA|SA|MA|Score
> |----|----|----|----|----|----|----|
> Original|84.85|81.82|85.99|79.63|48.67|-
> DPO_rej|33.33|41.41|67.46|62.04|49.19|59.32
> NPO|33.33|38.38|60.34|52.78|49.50|56.06
> ||
> GA_ret|33.33|34.37|76.94|66.28|53.95|65.70
> KLUE+GA_ret (ours)|33.33|36.36|83.41|74.54|57.48|69.76
> ||
> RMU|33.33|46.46|78.47|67.83|52.98|62.68
> KLUE+RMU (ours)|33.33|42.42|77.59|64.35|54.99|63.62
>
> $\\\\$
>
> **Llama 3.2 (3B)**
>
> |Method|UA|UA$^{‡}$|TA|SA|MA|Score
> |----|----|----|----|----|----|----|
> Original|87.88|90.91|87.28|85.65|50.57|-
> DPO_rej|30.30|46.46|69.61|55.56|54.30|58.25
> NPO|33.33|41.41|53.75|47.81|53.83|53.49
> ||
> GA_ret|33.33|48.48|68.10|57.87|53.89|57.84
> KLUE+GA_ret (ours)|33.33|46.46|77.16|64.81|54.04|62.38
> ||
> RMU|33.33|37.37|78.45|52.78|58.93|63.19
> KLUE+RMU (ours)|30.30|38.38|80.60|61.11|62.65|66.49
>
>
> In conclusion, KLUE enhances the unlearning abilities for various metrics even if it is applied to RMU.
>
> $\\\\$
>
> ### **3. Comparison with Harry Potter books [4] (Question # 1)**
>
> Our work focuses on evaluating the faithfulness of unlearning methods. From an unlearning research perspective, it is very easy to forget the text itself (even Harry Potter books). For example, we can destroy the model by preventing it from generating natural outputs.
>
> However, preserving other knowledge after forgetting a specific text is highly challenging. Additionally, ensuring the generalizability of unlearning is also difficult. Therefore, we decided to unlearn the simplest form of question (the Base QA form) and evaluate the generalizability of unlearning while preserving the model's other knowledge.
>
>
> $\\\\$
>
> If there are unsolved questions about the reasonability of our paper, we would appreciate being asked additional questions.
>
> $\\\\$
>
> [1] The WMDP Benchmark: Measuring and Reducing Malicious Use With Unlearning
>
> [2] https://github.com/centerforaisafety/wmdp
>
> [3] Negative Preference Optimization: From Catastrophic Collapse to Effective Unlearning
>
> [4] Who's Harry Potter? Approximate Unlearning in LLMs

---

> > ### Comment · Reviewer_Mrad · 2024-11-25
> >
> > Thanks for the explainations. After reading other reviewers' comments and the authors' responses, given the somewhat limited contribution of the paper, I decide to keep the score unchanged.

---

> > > ### Author Response · Authors · 2024-11-28
> > > **Response by the Author**
> > >
> > > Thank you for your response.
> > > Our study is the first to propose and argue that “there is an intersection between knowledge and that reliable unlearning should account for this intersection,” a perspective that has never been explored in previous unlearning research. Additionally, we introduce a novel unlearning method for accurately identifying neurons associated with specific privacy knowledge in a language model and selectively updating only these neurons—an approach not previously addressed in the field. We believe these contributions significantly advance the study of unlearning.
> > >
> > > We sincerely appreciate and respect the reviewers’ opinions. However, we kindly ask you to re-evaluate our submission, as we have diligently addressed and thoroughly discussed all the points raised by the reviewer.
> > >
> > > Thanks for your reading.

---

### Official Review · Reviewer_Bhk5 · 2024-11-01

**Soundness:** 2
**Presentation:** 3
**Contribution:** 2
**Rating:** 5
**Confidence:** 4

**Summary:**

To study the impact of machine unlearning on other related knowledge, the authors define a new concept called superficial unlearning. Based on the definition, they propose FaithUnBench to reveal that existing unlearning methods do not ensure faithful unlearning. To achieve faithful unlearning, the authors propose KLUE to update only knowledge-related neurons via the gradient ascent method.

**Strengths:**

1. It is interesting to evaluate the faithfulness of unlearning using Multi-hop QA and Same-answer QA.

2. It is reasonable to precisely update by localizing certain parameters to reduce the side effects of unlearning.

**Weaknesses:**

1.There is a lack of detailed comparison with existing datasets. For example, RWKU [1] also adopts 200 popular real world entities as the unlearning target knowledge and also considers the impact on related knowledge.

2.The proposed localization method is overly simplistic. How can neurons that express unrelated knowledge be avoided during localization? Additionally, neuron localization is not the only method for localizing key parameters. How does it compare with other localization methods?

3.There is a lack of more in-depth analysis. For example, in the analysis of the distribution of localized neurons, whether there is a certain distribution pattern for the neurons corresponding to different knowledge.

[1] RWKU: Benchmarking Real-World Knowledge Unlearning for Large Language Models

**Questions:**

1.There is a lack of quality assessment for the constructed benchmark. How can the noise introduced by GPT-4o be avoided?

2.Why can the problem of superficial forgetting be solved by neuron localization? Some case studies can be done on Multi-hop QA to check if MAf is really reduced rather than MAt being increased due to fewer updated model parameters.

---

> ### Author Response · Authors · 2024-11-20
> **Rebuttal by the Author (#1)**
>
> Thank you for the thorough feedback on our study. We will offer explanations, individually addressing the queries and questions you hold.
>
> $\\\\$
>
> ### **1. Detailed comparisons with existing datasets**
>
> In the "Rebuttal for All Reviewers," we discuss the characteristics of our benchmark in depth (at the top).
>
> $\\\\$
>
> ### **2. The positive impact of neuron localization (Weakness # 2 & Questions # 2)**
>
> **2-1. How did KLUE avoid selecting unrelated neurons?**
>
> The attribution score (gradient × activation) quantifies the relevance of neurons in predicting a text output, as it represents their contribution to the prediction [1]. Activation refers to the feature value, while the gradient indicates the direction and magnitude of the activation's influence on the output logit. Therefore, the attribution score can be used to compute a neuron's contribution to the output logit. As demonstrated by our experimental results (Section 5.3), using this score to identify knowledge-related neurons helps avoid unrelated knowledge when localizing neurons.
>
> $\\\\$
>
> **2-2. How did KLUE enable contextual unlearning?**
>
> To address the shortcomings of neuron localization, KLUE incorporates 'Superficial Knowledge Regularization' to mitigate shortcut unlearning, which merely reduces output text probability regardless of the given context. Our method enables contextual unlearning and improves the performance of MA$_{f}$ by excluding neurons associated with other unrelated knowledge during the unlearning process.
>
> We also present additional case studies for KLUE on Gemma-2 (2B).
>
> $\\\\$
>
> |Case|Type|Question|Logit before unlearning|Logit after unlearing|Pred before unlearning|Pred after unlearning
> |----|----|----|----|----|----|----|
> #1|Forget set question (UA)|What is the country of citizenship of Ellen DeGeneres?|0.5760|0.3107|United States of America|Austria|United States of America
> ||Multi-hop unlearn (MA$_f$)|Which city serves as the capital of the country that Ellen DeGeneres is a citizen of?|0.5759|0.3333|Washington D.C.|Amsterdam|Washington D.C.
> ||Multi-hop test (MA$_t$)|What is the profession of Ellen DeGeneres' mother?|0.4817|0.5701|columnist|columnist|columnist
> ||
> #2|Forget set question (UA)|What is Gwyneth Paltrow's country of citizenship?|0.5760|0.2123|United States of America|England|United States of America
> ||Multi-hop unlearn (MA$_f$)|What currency is associated with the country of citizenship of Gwyneth Paltrow?|0.5757|0.3334|United States dollar|Pound|United States dollar
> ||Multi-hop test (MA$_t$)|Which continent was Gwyneth Paltrow born in?|0.5758|0.5753|North America|North America|North America
> ||||||||
>
> $\\\\$
>
> **2-3. Why did we select 'attribution' to identify knowledge neurons?**
>
> We investigated several neuron localization methods [2,3] for detecting knowledge-relevant neurons in language models. However, these methods have only been evaluated on classification tasks and have not been applied to text generation tasks. The 'Skill Neurons' [2] method requires extensive computation to adapt it for text generation tasks, as it necessitates evaluating every possible word-piece combination to determine the knowledge neurons' distinguishing capabilities. Similarly, the 'Model Grafting' [3] method requires an additional training process to create masking parameters (used to identify knowledge neurons) equal in number to the model's parameters. Due to these implementation challenges of existing methods, we selected 'attribution' as our knowledge neuron detection method.
>
> $\\\\$
>
> [1] Mitigating Biases for Instruction-following Language Models via Bias Neurons Elimination
> [2] Finding Skill Neurons in Pre-trained Transformer-based Language Models
> [3] Task-Specific Skill Localization in Fine-tuned Language Models

---

> ### Author Response · Authors · 2024-11-20
> **Rebuttal by the Author (#2)**
>
> ### **3. The distribution of localized neurons (Weakness # 3)**
>
> We agree that conducting more examinations of the localized neuron distribution would enhance the quality of our paper. To address this, we have further analyzed the distribution of localized neurons. We compute attribution scores for each question and derive Jarcard and Cosine similarity between the attribution scores, where Jarcard similarity is computed using the identified knowledge neurons set, and Cosine similarity is computed using the attribution score distribution. We consider five types of questions for each target question: (1) Paraphrased question, (2) Same-entity question (same person but different contextual question), (3) Multi-hop question, (4) Same answer different context question, and (5) Unrelated question (randomly sampled).
> We sampled 402 questions, which included all these five types of questions, and computed the mean and standard deviations for the derived similarities. The experimental results are shown in the table below:
>
> $\\\\$
>
> ||Paraphrased question|Same-entity question|Multi-hop question|Same answer different context question|Unrelated question
> |----|----|----|----|----|----|
> Jacard Similarity|0.5570 (±0.04)|0.2453 (±0.03)|0.2631 (±0.04)|0.4386 (±0.07)|0.2406 (±0.03)
> Cosine Similarity|0.9467 (±0.02)|0.7194 (±0.07)|0.7441 (±0.07)|0.8873 (±0.05)|0.7319 (±0.06)
> |||||||
>
> $\\\\$
>
> As shown in the table, “paraphrased questions” show the highest similarity. Surprisingly, 'same-answer different context questions' show high similarity, although they are in a different context; the result shows that knowledge is largely determined from output texts. This result clearly justifies the necessity of our evaluation process using 'Same-answer QA Evaluation'.
>
> In addition, 'same-entity questions' and 'unrelated questions' show the lowest similarities. These results show that questions sharing the same entity are regarded as completely different knowledge if they are in different contexts (e.g., "Where was Barack Obama born?" vs. "What country does Barack Obama hold citizenship in?").
>
> $\\\\$
>
> ### **4. The Quality Assessment for the GPT-4o (Question # 1)**
>
> We adopt the question generation process following MQUAKE [1] since natural language question generation using ChatGPT variants is a kind of common and powerful method to generate questions. Even existing unlearning benchmarks, MUSE, KnowUnDo, TOFU, and RWKU, also adopt ChatGPT variants for generating natural language questions.
>
> However, We have fully acknowledged that additional investigation should be conducted to ensure the dataset quality. Therefore, we conducted a human evaluation to investigate the quality of the generated questions. Specifically, we recruited crowd workers fluent in English through the university’s online community and let them investigate 800 generated natural language questions (10% of the entire questions). The results revealed 0% of the error rate, ensuring the quality of our constructed benchmark.
>
> $\\\\$
>
> If there are remaining unsolved questions about the reasonability of our benchmark and method, we would appreciate being asked additional questions.
>
> $\\\\$
>
> [1] MQuAKE: Assessing Knowledge Editing in Language Models via Multi-Hop Questions

---

> > ### Comment · Reviewer_Bhk5 · 2024-11-28
> >
> > Thanks for the responses. I have decided to maintain the rating score.

---

### Official Review · Reviewer_FJiB · 2024-11-01

**Soundness:** 3
**Presentation:** 3
**Contribution:** 2
**Rating:** 6
**Confidence:** 4

**Summary:**

The authors define superficial unlearning and construct a new benchmark, FaithUnBench, to analyze and achieve faithful unlearning.
Furthermore, the authors propose a novel knowledge-localized unlearning method, KLUE, to mitigate superficial unlearning and
reveal that our method outperforms other unlearning methods, dramatically mitigating superficial
unlearning.

**Strengths:**

This article contributes a dataset and an effective method to the community.

**Weaknesses:**

1. There is no comparison of the proposed dataset with previous datasets, such as MUSE, WMDP, KnowUnDo, TOFU.
2. In the Faithful Unlearning setting, some knowledge related to current entities should not be forgotten. Has this consideration been taken into account in the constructed dataset? For example, in Figure 1, changing Tom Cruise's nationality should not affect the answer regarding his notable works. Intuitively, the unlearning process is more likely to damage content related to Tom Cruise rather than affect another person's nationality.

**Questions:**

The goal of unlearning is to completely forget specific knowledge. Has the author considered the following scenario: testing whether the knowledge has truly been forgotten by asking about it in a different language?

---

> ### Author Response · Authors · 2024-11-20
> **Rebuttal by the Author**
>
> We sincerely appreciate your thorough examination of our paper. Your considerate evaluation and constructive feedback are truly valued. We will provide responses to the questions you have requested.
>
> $\\\\$
>
> ### **1. Detailed comparisons with existing datasets (Weakness # 1)**
>
> In the "Rebuttal for All Reviewers," we discuss the characteristics of our benchmark in depth (at the top).
>
> $\\\\$
>
> ### **2. Details of our benchmark in considering different knowledge of each entity as the independent evaluation component (Weakness # 2)**
>
> We generate multiple knowledge questions for each entity and consider them independent elements for evaluation. For example, suppose the question *"What is the country of citizenship of Tom Cruise?"* is included in the forget set. Then, another question *"Where was Tom Cruise born?"* can be included in the test set (for measuring TA and MA scores). We adopt this framework to consider a more general use case of unlearning, differentiating from existing benchmarks like TOFU [1] (TOFU aims to forget the entire knowledge of each entity). In Table 6 of our paper (Appendix), we present that each unlearning cluster is constructed only on one knowledge question of each entity (not all knowledge).
>
> $\\\\$
>
> ### **3. Evaluating the unlearned knowledge about different languages**
>
> We translate our dataset into French and German to determine whether KLUE can completely erase the knowledge in different language settings. Specifically, we translate the forget set (5\%) into French and German for the experiments.
> The below table shows the examples of translated questions.
>
> |Original | French | German
> |----|----|----|
> Who is the mother of Michelle Obama? | Qui est la mère de Michelle Obama? | Wer ist die Mutter von Michelle Obama?
> What is Kim Kardashian's religion? | Quelle est la religion de Kim Kardashian?| Was ist die Religion von Kim Kardashian?
> Who is the father of Hillary Clinton? | Qui est le père de Hillary Clinton? | Wer ist der Vater von Hillary Clinton?
> Where was Leonardo DiCaprio born? | Où est né Leonardo DiCaprio? | Wo wurde Leonardo DiCaprio geboren?
> Who was Theodore Roosevelt's father? | Qui était le père de Theodore Roosevelt? | Wer war der Vater von Theodore Roosevelt?
> ...|...|...
>  | | |
>
> We unlearn the Gemma-2 (2B) model using KLUE for the questions of the original language (English) and evaluate the unlearned model in translated questions (French and German).
> The table below shows the experimental results of the French and German questions. The results reveal that KLUE has a generalization ability to other languages.
>
> || Before unlearning | After unlearning
> |----|----|----|
> English | 84.85 | 33.33
> French | 81.82 | 45.45
> German | 84.85 | 48.48
>  | | |
>
> $\\\\$
>
> [1] TOFU: A Task of Fictitious Unlearning for LLMs

---

> ### Comment · Reviewer_FJiB · 2024-11-27
> **keep the score as it is.**
>
> Thank you for your response. After careful consideration, I’ve decided to keep the score as it is.

---

### Official Review · Reviewer_c7Yt · 2024-11-04

**Soundness:** 3
**Presentation:** 3
**Contribution:** 2
**Rating:** 6
**Confidence:** 3

**Summary:**

This paper focuses on the issue of "unfaithful" unlearning knowledge from the LLMs, including failing to erase the knowledge it should remove and unintentionally erasing irrelevant knowledge. A new benchmark, FaithUnBench, is proposed for analyzing and evaluating the faithfulness of unlearning in the knowledge QA settings, consisting of Paraphrased QA, Multi-hop QA, and Same-answer QA datasets. The authors also present an approach to mitigate the issue. In particular, it identifies and updates only the knowledge-related neurons based on selected unforgotten samples.

**Strengths:**

* Present a dataset for faithful unlearning knowledge from LLMs, targeting at different categories of unfaithful issues
* Experimental results demonstrate the effectiveness of the approach, with further detailed analysis

**Weaknesses:**

* The approach relies on the world knowledge graph, which is restricted to the triple-based QA settings

**Questions:**

* Have you tried larger p value for the experiments in section 5.6? It seems that the tendency is still going up.
* Have you compared different prompting templates for MCQA? Furthermore, apart from the convenience for evaluation, have you compared it with other ways of prompting?

---

> ### Author Response · Authors · 2024-11-20
> **Rebuttal by the Author**
>
> Thank you for taking the time to review our paper so thoroughly. We greatly value your thoughtful insights and constructive suggestions. We will ensure that your questions are addressed in our response.
>
> $\\\\$
>
> ### **1. Detailed comparisons with existing datasets**
> In the "Rebuttal for All Reviewers", we discuss the characteristics of our benchmark in depth (at the top).
>
> $\\\\$
>
> ### **2. The justification for using the knowledge graph (Weakness # 1)**
>
> Most questions can be expressed in the knowledge graph; thus, our setting can cover an extensive area of real-world knowledge.
>
> $\\\\$
>
> ### **3. Larger neuron ratio (p) experiments (Question # 1)**
>
> We conduct the experiments on a larger neuron ratio to investigate the KLUE method further for Gemma-2 (2B).
>
> |Neuron ratio ($p$) |UA|UA$^{‡}$|TA|SA|MA|Score
> |----|----|----|----|----|----|----|
> 0.01 | 33.33 | 42.42 | 81.03 | 68.98 | 56.33 | 65.98
> 0.05 | 33.33 | 36.36 | 83.41 | 74.54 | 57.48 | 69.76
> 0.1 | 33.33 | 37.37 | 83.62 | 74.54 | 55.50 | 69.07
> 0.2 | 33.33 | 42.42 | 81.09 | 67.13 | 57.40 | 65.8
> 0.5 | 33.33 | 39.39 | 82.97 | 72.69 | 58.81 | 68.77
>
> We reveal that even the larger ratios show comparable results, however, simply increasing the neuron ratio does not enhance the performance.
>
> $\\\\$
>
> ### **4. Various prompting experiments (Question # 2)**
>
> We fully acknowledge that the quality of our research will be enhanced if we conduct additional experiments using various prompts.
>
> $\\\\$
>
> **Different prompt templates**
>
> We first paraphrase the instruction of the utilized prompt template (Appendix B.1) and create five new instructions as follows:
>
> > 1. *Pick the appropriate option for the question from the provided options. You should answer without further explanation.*
> > 2. *Select the correct answer for the given question from the options. Write only the word without explanation.*
> > 3. *Answer the given question by choosing the appropriate answer from the given options. Do not include any explanations.*
> > 4. *Select the correct answer to the following question among the options. Only the exact word should be written, with no explanation.*
> > 5. *Select the proper answer to the question from among the given options. Write only the exact word without any additional explanation.*
>
> We conduct the evaluation on these newly selected instructions using the unlearned Gemma-2 2B KLUE model, and the experimental results are shown below:
>
> |Prompt idx|UA|UA$^{‡}$|TA|SA|MA|Score
> |----|----|----|----|----|----|----|
> original | 33.33 | 36.36 | 83.41 | 74.54 | 57.48 | 69.76
> 1 | 39.39 | 37.37 | 82.76 | 73.61 | 57.16 | 69.04
> 2 | 39.39 | 42.42 | 81.47 | 73.61 | 57.51 | 67.54
> 3 | 36.36 | 38.38 | 83.41 | 74.54 | 58.10 | 69.42
> 4 | 36.36 | 38.38 | 83.41 | 74.54 | 57.21 | 69.20
> 5 | 39.39 | 38.38 | 82.33 | 76.39 | 56.55 | 69.22
> | | | | | |
>
> From the experiments, we reveal that the newly adopted prompts perform similarly to the original prompt.
> Their performance on the UA score is slightly higher than the original one since we early stopped the unlearning process based on the UA score evaluation for the original prompt.
>
> $\\\\$
>
> **Other ways of prompting**
>
> We also evaluate the original Gemma-2 (2B) and the KLUE unlearned Gemma-2 (2B) models without any instruction. Specifically, we input only a question (e.g., *"Who is the mother of Michelle Obama?"*) to the model without using any instruction or prompt.
> We measure the performance using ROUGE-L, and the experimental results are shown in the below:
>
> ||UA|UA$^{‡}$|TA|SA|MA|Score
> |----|----|----|----|----|----|----|
> Before unlearning | 84.85 | 82.40 | 86.14 | 80.56 | 48.71 | 58.25
> After unlearning | 38.82 | 43.84 | 83.97 | 76.99 | 57.00 | 68.53
> | | | | | |
>
>
> Of course, ROUGE-L has its limitations, as it does not account for exact matching. However, its evaluation results are highly correlated with MCQA outcomes and can serve as an alternative evaluation metric.

---

### Author Response · Authors · 2024-11-20
**Rebuttal for All Reviewers**

We sincerely thank all the reviewers for their invaluable comments and suggestions. Your thoughtful feedback and efforts have greatly improved our work, and we deeply appreciate your time and guidance.

All the contents addressed in our rebuttal process will be included in the final version of our paper.

$\\\\$

### **Detailed comparisons with existing datasets**

Our benchmark aims to unlearn real-world entity knowledge, which can be prevalent in various language models, to consider the most practical situation of knowledge unlearning. Furthermore, our benchmark deals with the complex and interconnected nature of world knowledge; thus, we introduce three types of unlearning evaluation aspects (Paraphrased QA, Multi-hop QA, and Same-answer QA) for more deep analysis of real-world knowledge unlearning. We propose detailed comparisons with existing datasets to clearly show the novelty of our benchmark. We can summarize the differences in our benchmark in the table below.

$\\\\$

||MUSE [1]|KnowUnDo [2]|WMDP [3]|TOFU [4]|RWKU [5]|FaithUn (Ours)
|----|----|----|----|----|----|----|
Knowledge Source|BBC News & Harry Potter book|Copyrighted books|Hazardous knowledge|Fictitious author|Real-world Entity|Real-world Entity
\# Unlearning Entities|N/A|N/A|N/A|200|200|200
\# Forget Probes|889|987|4,157|4,000|13,131|8,377
Knowledge Exists in LLMs|X|X|O|X|O|O
Related Knowledge|X|X|X|X|O|O
Paraphrased QA Evaluation|X|X|X|X|O|O
Multi-hop QA Evaluation|X|X|X|X|X|O
Same-answer QA Evaluation|X|X|X|X|X|O

$\\\\$

In summary, only RWKU and our benchmark address real-world entities as targets for unlearning. Additionally, MUSE, KnowUnDo, and TOFU require fine-tuning to inject knowledge before unlearning, which may reduce their practicality. Furthermore, most existing benchmarks, except for RWKU and our benchmark, have not considered related knowledge.

However, RWKU has not dealt with "multi-hop QA evaluation", which assesses the interconnections between knowledge, and "same-answer QA evaluation", which assesses whether unlearning algorithms degrade output probabilities without considering the given contexts. For example, RWKU includes an unlearning target text, *"Please forget Stephen King, who is a American author, renowned as the 'King of Horror'."*, and also contains a related knowledge question, *"Who plays the character Jack Torrance in the film 'The Shining'?"*. The two questions are quite related, but they are not completely interconnected like multi-hop questions.

In conclusion, the main contribution of our benchmark lies in evaluating whether unlearning methods perform **faithful unlearning** while considering **knowledge interconnection** within the **real-world entity unlearning** setting.

$\\\\$

[1] MUSE: Machine Unlearning Six-Way Evaluation for Language Models

[2] To Forget or Not? Towards Practical Knowledge Unlearning for Large Language Models

[3] The WMDP Benchmark: Measuring and Reducing Malicious Use With Unlearning

[4] TOFU: A Task of Fictitious Unlearning for LLMs

[5] RWKU: Benchmarking Real-World Knowledge Unlearning for Large Language Models

---

### Meta-Review · Area_Chair_AduD · 2024-12-20

**Metareview:**

This paper addresses the issue of 'unfaithful' unlearning in large language models (LLMs), which includes both the failure to remove intended knowledge and the unintended erasure of irrelevant knowledge. The authors propose a new benchmark, FaithUnBench, designed to evaluate the faithfulness of unlearning in knowledge-based QA settings. This benchmark consists of three datasets: Paraphrased QA, Multi-hop QA, and Same-answer QA. Additionally, the paper presents an approach to mitigate the problem, which involves identifying and updating only the knowledge-related neurons based on selected unforgotten samples. The paper does not provide a detailed comparison of the proposed FaithUnBench dataset with previous datasets like MUSE, WMDP, KnowUnDo, TOFU, and RWKU. The proposed neuron localization method is a bit incremental. There is concern about how the method avoids neurons expressing unrelated knowledge. The paper lacks a more thorough analysis, particularly regarding the distribution patterns of localized neurons. It would be valuable to explore whether neurons corresponding to different knowledge types follow a specific distribution pattern.

**Additional Comments On Reviewer Discussion:**

After discussion, some reviewers still believe that this paper has room that could be improved.

---

### Decision · Program_Chairs · 2025-01-22

Reject